# Prevalence of Intestinal Parasitic Infections, Genotypes, and Drug Susceptibility of *Giardia lamblia* among Preschool and School-Aged Children: A Cross-Sectional Study in Thailand

**DOI:** 10.3390/tropicalmed8080394

**Published:** 2023-08-01

**Authors:** Boonchai Wongstitwilairoong, Thunyarat Anothaisintawee, Nattaya Ruamsap, Paphavee Lertsethtakarn, Paksathorn Kietsiri, Wirote Oransathid, Wilawan Oransathid, Siriphan Gonwong, Sasikorn Silapong, Umaporn Suksawad, Siriporn Sornsakrin, Ladaporn Bodhidatta, Daniel M. Boudreaux, Jeffrey R. Livezey

**Affiliations:** Department of Bacterial and Parasitic Diseases, US Army Medical Directorate of the Armed Forces Research Institute of Medical Sciences, Bangkok 10120, Thailand; boonchaiw@afrims.org (B.W.); paphaveel@afrims.org (P.L.); paksathornk@afrims.org (P.K.); wiroteo@afrims.org (W.O.); wilawano@afrims.org (W.O.); siriphang@afrims.org (S.G.); sasikorns@afrims.org (S.S.); umaporns@afrims.org (U.S.); siriporns@afrims.org (S.S.); ladaporn@hotmail.com (L.B.); daniel.boudreaux.mil@afrims.org (D.M.B.); jeffrey.livezey.mil@afrims.org (J.R.L.)

**Keywords:** intestinal parasite, *Giardia lamblia*, prevalence, drug susceptibility, genotype, cross-sectional study

## Abstract

This study aimed to estimate the prevalence of intestinal parasitic infections in children and assess the drug susceptibility and genotypes/assemblages of *Giardia lamblia* in Thailand. This cross-sectional study was conducted among children aged 3–12 years in Sangkhlaburi District, Kanchanaburi Province, Thailand, between 25 September 2017 and 12 January 2018. Parasites were identified by stool microscopic examination, cultivation of intestinal parasitic protozoa, and enzyme-linked immunosorbent assay (ELISA). Drug susceptibility and genotype of *G. lamblia* were performed, respectively, by a resazurin assay and Triosephosphate Isomerase A and B genes using modified primers and probes. Among the 661 participants, 445 had an intestinal parasitic infection, resulting in a prevalence of 67.32% (95% CI: 63.60–70.89%). *Blastocystis hominis* was the most prevalent protozoa infection (49.32%; 95% CI: 45.44–53.22%), while *Ascaris lumbricoides* was the most prevalent helminth infection (0.91%; 95% CI: 0.33–1.97%). The prevalence of *G. lamblia* was 17.40%, with genotype B being the most common. According to our study, intestinal parasitic infections were commonly found in Thai children. *G. lamblia* was the most common pathogenic protozoa infection identified and exhibited less susceptibility to metronidazole compared to furazolidone and mebendazole.

## 1. Introduction

Intestinal parasitic infections are a major public health problem, leading to substantial morbidity and mortality among preschool and school-age children in developing countries [1,2]. Tropical and subtropical regions have the highest prevalence of intestinal parasitic infections, primarily due to deprived socioeconomic status and poor personal and environmental hygiene among the majority of the population [3,4]. Despite having a low mortality rate, intestinal parasitic infections have long-lasting impacts on the host’s health and nutritional state [5]. These parasites induce intestinal damage, leading to a reduction in the absorption of nutrients [3]. Furthermore, symptoms caused by the parasites, such as diarrhea, vomiting, and loss of appetite, decrease food consumption and hinder the absorption of fats and proteins from the gastrointestinal tract [6]. These consequences result in malnutrition and nutrient insufficiency in infected individuals.

Despite Thailand’s improvements in infrastructure and public utilities, the water supply system continues to be insufficient in many regions. People in remote areas still use groundwater or other untreated natural water sources [7], exposing them to a heightened risk of intestinal parasitic infections, especially in children. Based on findings from previous studies [8,9,10,11], the prevalence of intestinal parasitic infections in school-aged children in Thailand ranged from 12.6% to 68.0%, depending on the study region. *Ascaris lumbricoides*, *Trichuris trichiura*, and hookworms were the most prevalent helminths, while *Giardia lamblia* and *Entamoeba coli* were the most prevalent protozoa. Despite the implementation of treatment and mass drug therapy targeting school children in Thailand to reduce the prevalence of parasitic infection, several regions continue to report a persistently high prevalence of intestinal parasites in schoolchildren [10,12]. This is believed to be due to medication noncompliance in infected patients and drug resistance [13].

Currently, eight assemblages (A–H) of *G. lamblia* have been identified, with assemblages A and B being associated with human infections with potential zoonotic infections. In contrast, other assemblages are associated with different hosts [14]. There are few data available on the current state of *G. lamblia* in Thailand [15,16]. The inclusion of *G. lamblia* detection and genotyping can provide valuable epidemiological information and potentially enhance our understanding of the *G. lamblia* transmission route. For *Giardia* therapy, several classes of antimicrobial drugs are available for treating giardiasis, however, treatment failure is increasingly reported, especially for the most commonly used drugs, such as metronidazole and tinidazole, which are members of the nitroimidazole family [17]. Potential factors such as inappropriate dosing and repeated courses of treatment over long periods may also contribute to the development of drug resistance in *Giardia* [18]. Numerous reports have demonstrated an increasing resistance or tolerance to anti-parasitic agents that are commonly used, especially for 5-nitroimidazoles (e.g., metronidazole and tinidazole) which are the drug of choice for *Giardia* therapy [19]. Particularly, therapeutic failure after treatment with 5-nitroimidazole compounds (e.g., metronidazole, secnidazole, and tinidazole) has been frequently reported [20,21,22]. Moreover, up to 50% of nitroimidazole treatment failure in giardiasis has been identified among travelers returning from South Asia [23].

Therefore, this cross-sectional study aimed to estimate the prevalence of intestinal parasitic infections and assess the association between intestinal parasitic infections and nutritional status in children living in the Sangkhlaburi district, Kanchanaburi province Thailand. Moreover, genotypes and in-vitro drug susceptibility of *G. lamblia* were also evaluated to monitor genetic diversity and drug susceptibility profiles of *Giardia* circulating in the endemic settings of Thailand.

## 2. Materials and Methods

### 2.1. Study Sites and Study Population

This cross-sectional study was conducted in Sangkhlaburi District, Kanchanaburi Province, Thailand, between 25 September 2017 and 12 January 2018. Children aged 3–12 years old who attended Bann Huay Malai, Border Patrol Police United Bank Bangkok, and United Christian schools located in Sangkhlaburi District were enrolled in this study. Figure 1 shows a map with the geographic location of the study sites and the province of the study.

Potential participants were recruited by invitation. Parents/guardians of the potential subjects were informed verbally about the study procedures by research nurses. The parents/guardians were requested to sign an informed consent form to allow the children to participate in the study. A signed assent form was also obtained from children aged 7–12 years old who were willing and had consent from their parents/guardians to participate in the study. Children taking anti-parasitic drugs in the past 2 weeks prior to enrollment were excluded from the study.

This study was approved by the Ethical Review Committee of the Ministry of Public Health, Nonthaburi, Thailand, and the Walter Reed Army Institute of Research, Silver Spring, Maryland.

### 2.2. Data Collection

After obtaining signed assent/consent forms, demographic (i.e., age, sex) and clinical data, including stool grade and symptoms (e.g., abdominal cramps, nausea/vomiting, fever) were collected by interviewing the study’s subjects for children aged 7–12 years and by interviewing their parents/guardians for children aged less than 7 years. Stool grade was classified into normal stool (fully formed or soft stool) and diarrheal stool (thick liquid, opaque watery, or rice watery stool).

### 2.3. Nutritional Assessment

The body weight of the study’s participants was measured to the nearest 0.1 kg using an electronic digital scale. Their height was determined to the nearest 0.1 cm. Their Body Mass Index (BMI) was calculated by dividing body weight (in kg) by height (in meter^2^). The participant’s nutritional statuses were defined according to the World Health Organization (WHO) criteria [24]. The participants were classified as wasting, stunting, and underweight if their weight for height (WFH), height for age (HFA), and weight for age (WFA) were lower than −2 standard deviation (SD), respectively.

### 2.4. Stool Collection

Single stool samples were collected by self-collection or with assistance from study personnel or parents/guardians. Parasitic identification was performed using 3 methods: (1) stool microscopic examination, (2) cultivation of intestinal parasitic protozoa, and (3) enzyme-linked, immunosorbent assay (ELISA) [25]. Any stool specimens that were *Giardia* positive by microscopic examination and/or ELISA were selected for cyst purification and excystation. Trophozoite isolates were used for drug susceptibility testing.

### 2.5. Laboratory Evaluations

#### 2.5.1. Microscopic Examination

Fresh stool samples were examined microscopically by trained technicians at study sites on a direct smear for fecal white blood cells (WBC), fecal red blood cells (RBC), intestinal protozoa, and helminths [26].

#### 2.5.2. Cultivation of Intestinal Parasites

Approximately 50 mg of the stool sample was cultivated in a screw-capped tube containing Boeck and Drbohlav Locke egg medium (LE media) [27] and incubated at 37 °C for at least 72 h prior to examination. A drop of sediment was extracted from the tube and examined on a microscope slide. If no parasite was identified from the first examination, the sediment was re-suspended and transferred to a fresh LE culture tube for an additional 72 h of the incubation period for a second examination. If no parasite was seen from the second examination, sub-culturing was performed as described for the second examination with an additional 72 h of incubation. The culture was classified as negative if no parasite was identified from the third examination.

#### 2.5.3. Identification of *Entamoeba histolytica*, *G. lamblia*, and *Cryptosporidium* by ELISA

An aliquot of each stool specimen (approximately 0.5–2 mL) was qualitatively detected *E. histolytica*, *G. lamblia*, and *Cryptosporidium* by a sandwich ELISA using commercial diagnostic test kits (TechLab, Inc., Blacksburg, VA, USA) [25]. The procedure was performed according to the manufacturer’s instructions.

#### 2.5.4. Purification and Excystation of *G. lamblia* Cysts

The stool mixture (5 g in 30 mL of cold deionized water (CDW)) was filtered through a sterile gauze. The filtrate was layered on 0.85 M sucrose and centrifuged at 600× *g*, 10 min, 4 °C. Interface was collected and then layered on a discontinuous sucrose gradient (0.85 M and 0.4 M). After harvesting the interface, cysts were washed and suspended in CDW with antibiotics (250 units/mL of penicillin G, 250 µg/mL of streptomycin, 100 µg/mL of gentamicin, and 0.25 µg/mL of amphotericin B; all reagents from Invitrogen, Carlsbad, CA, USA). Two steps of excystation were followed according to the established procedures [28]. Details of the procedure are provided in Appendix B.

#### 2.5.5. Drug Susceptibility Testing of *G. lamblia*

Stocks of metronidazole, furazolidone, and mebendazole (Sigma-Aldrich, Burlington, MA, USA) were prepared by dissolving them in 100% Dimethyl sulfoxide (DMSO; Sigma Aldrich, Burlington, MA, USA). The maximal dilution of compounds did not exceed 0.1% (*v*/*v*) DMSO, which had no effect on *G. lamblia* growth. A resazurin assay [29] was performed to determine the drug’s susceptibility against metronidazole, furazolidone, and mebendazole. *G. lamblia* WB-C6 (ATCC #50803; genotype AI) was used as a reference isolate to identify the optimal growth conditions. Prior to testing drug susceptibility, growth conditions of *G. lamblia* WB-C6 were determined by varying cell seeding (500 to 2 × 10^4^ trophozoites/well) in 96-well plates and varying incubation times (24, 48, and 72 h). Optimal conditions were determined to be 1 × 10^4^ trophozoites/well and incubation of 48 h (see Appendix A).

To test drug susceptibility, 100 µL of the serially diluted drugs in TYI-S-33 medium were incubated with 100 µL of 1 × 10^4^ trophozoites in 96-well microplates (Corning Costar# 3603, Glendale, AZ, USA) at 37 °C under oxygen-deprived conditions for 48 h. After washing wells with 200 µL of phosphate-buffered-saline (PBS), 220 µL of 20 µM resazurin in PBS was added and then incubated for 4 h. Fluorescence intensity was measured at 595 nm using a 550 nm excitation wavelength (SpectraMax M2e microplate reader; Molecular Devices, Silicon Valley, CA, USA). In each plate, background (medium only), vehicle (0.1% DMSO), and no vehicle (*Giardia* and media) were included as control wells. Assays were run in triplicates with three independent experiments. The percentage of growth inhibition was calculated [30]. The half-maximal inhibitory concentrations (IC_50_) were determined from the generated dose-response curve (GraphPad Prism 7.05; GraphPad Software Inc., San Diego, CA, USA).

#### 2.5.6. *G. lamblia* Detection and Genotyping

Nucleic acids were extracted using the QIAamp Fast DNA stool kit as described previously [31] with an added step of using a bead beater. Approximately 10^6^ copies of phocine herpesvirus (PhHV) were spiked into the lysis buffer to serve as a DNA extrinsic control to monitor the efficiency of the extraction and inhibition in real-time PCR reactions [32].

Samples that were positive by ELISA and/or microscopy were subjected to *Giardia* screening with real-time PCR using modified primers and probes [31,33] (see Appendix A). The reaction was set up in 20 µL of IQ power mix (Bio-Rad, Hercules, CA, USA), 4 µL of DNA template, 0.2 µM of each primer, and 0.1µM of the probe. The thermal cycling condition was as follows: 95 °C for 5 min, followed by 45 cycles of 95 °C for 15 s, and then 60 °C for 1 min in RotorgeneQ (Qiagen, Germantown, MD, USA).

All positive samples were further genotyped based on Triosephosphate Isomerase A and B (TPI A and TPI B) genes using modified primers and probes [34] (see Appendix A). The reaction and thermal cycling conditions were similar to the screening assay described above, with the exception that the primer concentration was 0.25 µM.

### 2.6. Statistical Analysis

Characteristics of the study participants (i.e., demographic and clinical data), genotypes, and drug susceptibility of *G. lamblia* were described by frequency and percentage for categorical data. Continuous data were presented by mean and standard deviation (SD) if the data had a normal distribution; otherwise, the median and range were applied. The prevalence of intestinal parasitic infections in participants was estimated by dividing the number of children who had intestinal parasitic infections by the total number of participants. The characteristics of subjects that had and did not have intestinal parasitic infections were compared using the t-test for continuous data and the chi-square test for categorical data. The association between intestinal parasitic infections and nutritional status was assessed using logistic regression analysis. One-way ANOVA was used to compare the mean IC_50_ among the 10 *G. lamblia* isolates. One-way ANOVA with Tukey’s multiple comparisons test was used for the comparison of the mean IC_50_ among the three drugs in each isolate. A *p*-value less than 0.05 with a two-sided test was considered statistical significance for all tests. All statistical analyses were performed using STATA program version 17 (College Station, TX, USA).

## 3. Results

A total of 661 participants were included in this study. Around half of the study’s participants (49.17%) were recruited from the United Christian School, followed by Bann Huay Malai (34.19%) and the Border Patrol Police United Bank Bangkok School (16.64%). The characteristics of the study’s participants are presented in Table 1. The mean age, weight, and height of the participants were 8.29 years (SD: 2.14), 26.72 kg (9.56), and 126.54 cm (13.01), respectively. A total of 357 participants were female (54%). Only 27 (4%) and 36 (5%) participants had diarrhea symptoms and fever at the time of enrollment. None of the students had red blood cells in their stool, and only 9 (1%) had white blood cells in their stool. Comparisons of demographic data, stool characteristics, and clinical signs and symptoms between students with and without intestinal parasitic infections did not show any significant difference, except that the participants from the Border Patrol Police United Bank Bangkok School had a higher rate of intestinal parasitic infections (80%) than the participants from Bann Huay Malai (67.26%) and United Christian School (63.08%). Additionally, there were no significant differences observed in symptoms and signs between children infected with pathogenic intestinal parasites (i.e., *G. lamblia*, *E. histolytica*, and intestinal helminths) and children who were not infected with these parasites (see Appendix A).

### 3.1. Prevalence of Intestinal Parasitic Infection

The prevalence of intestinal parasitic infections among study participants is presented in Table 2. A total of 445 students had intestinal parasitic infections, with a prevalence of 67.32% (95% CI: 63.60–70.89%). The prevalence of intestinal parasitic infections was highest in the Border Patrol Police United Bank Bangkok School (80%), followed by Bann Huay Malai School (67.26%) and United Christian School (63.08%). Protozoa, helminth, and both protozoa and helminth infections were present in 430, 2, and 13 of the students, respectively, indicating a prevalence of 65.05, 0.3%, and 1.97%, respectively. The prevalence of protozoa and helminth infections was highest in the Border Patrol Police United Bank Bangkok School (80% for protozoa infections, 2.73% for helminth infections), followed by the Bann Huay Malai School (67.02% for protozoa infections, and 2.65% for helminth infections), and the United Christian School (62.77% for parasitic infections and 1.85% for helminth infections). When conducting a stratified analysis based on age groups (preschool and school-aged), it was observed that helminthic infections were more prevalent among preschool children, whereas protozoa infections were commonly found in school-aged children (see Table 2).

Seven intestinal protozoa were identified, *B. hominis* being the most prevalent, followed by *Endolimax nana*, *G. lamblia*, *E. coli*, *Iodamoeba buetschlii*, *Pentatrichomonas hominis*, and *E. histolytica*. A total of 20 (3.03%) of the study’s participants were infected with unspeciated *Entamoeba* (*Entamoeba* spp.). None of the study’s participants were infected with *Cryptosporidium*. Pathogenic protozoa like *G. lamblia* were found to be more prevalent in preschool children compared to schoolchildren (see Table 2).

The prevalence of all helminths detected in the study’s subjects was lower than 1%. *A. lumbricoides* was the most prevalent, while *S. stercoralis* and *T. trichiura* had the lowest prevalence. The prevalence of *A. lumbricoides* and *Enterobius vermicularis* was higher in preschool children compared to schoolchildren.

### 3.2. Association between Intestinal Parasitic Infection and Nutritional Status

Eighty-seven, 89, and 50 students had WFA, HFA, and WFH under 2 SD, repsectively. Thus, the prevalence of underweight, stunting, and wasting in participants was 13.16% (95% CI: 10.68–15.98%), 13.46% (95% CI: 10.95–16.31%), and 7.56% (95% CI: 5.67–9.85%), respectively. Students who were infected with protozoa and/or helminths were more likely to be underweight, stunting, and wasting. These associations, however, failed to reach statistical significance. However, when assessed, the association between infections from the most prevalent protozoa (i.e., *G. lamblia*, *B. hominis*, and *E. nana*) and nutritional status, infections with these protozoa were significantly associated with children being underweight (see Table 3).

While comparing individual parasites, *B. hominis* was significantly associated with being underweight, while *E. nana* and *I. buetschlii* infections were significantly associated with a student being stunted (see Table 3). Infections of *A. lumbricoides* and/or *E. vermicularis* were highly associated with being underweight, stunted, and wasted in the students, but these associations were not statistically significant. 

### 3.3. Excystation of G. lamblia Cysts

A total of 115 stools were positive for *G. lamblia* by either microscopy, ELISA, or both. An amount of 85 stools that were positive by microscopy and/or strongly positive by ELISA (optical density of >1.5) were selected for cyst purification.

The cyst purification yielded 61 specimens that contained a significant number of purified cysts (at least 2 × 10^4^ cysts). Of the 61 stool specimens, *G. lamblia* trophozoites with axenic cultures were successfully established in 10 (16%) specimens (see Appendix A). Surprisingly, a significant number of purified cysts (51/61 stools; 83.6%) were not successfully established as cultures. The purified cysts from 5 stool specimens (5/61; 8.2%) were heavily contaminated with bacteria during the excystation process, while cysts from 46 stool specimens (46/61; 75.4%) were capable to excyst into motile trophozoites but failed to attach to the tube wall. We observed that a high number of purified cysts in stool samples is one of the critical factors for successful excystation and axenization. Therefore, only 10 *G. lamblia* isolates were tested for drug susceptibility.

### 3.4. Drug Susceptibility Test of G. lamblia Isolates

The IC_50_ value for each *G. lamblia* isolate was assessed after 48 h of drug exposure (Figure 2). For metronidazole, the IC_50_ varied from 0.484 to 2.1 μM, representing a 4.3-fold variation of susceptibility. For furazolidone, the IC_50_ varied from 0.079 to 0.473 μM, representing a 6-fold variation of susceptibility. For mebendazole, the IC_50_ varied from 0.030 to 0.114 μM, representing a 3.8-fold variation of susceptibility. By statistical analysis, the mean IC_50_ values showed significant difference across the 10 *G. lamblia* isolates for mebendazole (*p*-value = 0.0127), and a strongly significant difference was detected for metronidazole and furazolidone (*p*-value < 0.0001).

The mean IC_50_ (±SD) of all 10 isolates were 1.18 ± 0.55 μM for metronidazole, 0.17 ± 0.11 μM for furazolidone, and 0.07 ± 0.03 μM for mebendazole. In these in vitro tests, our *G. lamblia* isolates had the least susceptibility to metronidazole among the three drugs tested. Further, the ratio of drug concentration for metronidazole vs. mebendazole was 17-fold (1.18/0.07), while metronidazole vs. furazolidone was 7-fold (1.18/0.17). We also found that IC_50_ of the majority of *G. lamblia* isolates tested against metronidazole were significantly higher than that tested against mebendazole (10/10 isolates; *p*-values ranging from <0.0001 to 0.0009) and tested against furazolidone (9/10 isolates; *p*-values ranging from <0.0001 to 0.0018). Apparently, the mean IC_50_ of one sample against furazolidone was significantly higher than mebendazole (*p*-value = 0.0009). However, the remaining 9 samples did not show a significant difference in IC_50_ between these two drugs. Collectively, our in vitro findings suggest that metronidazole was considered less effective as a giardiasis treatment than mebendazole and furazolidone.

### 3.5. G. lamblia Genotypes

Among 115 stool samples that were positive for *G. lamblia*, eight stool samples were positive from microscopy alone, 55 samples were positive from ELISA alone, and 52 samples were positive from both ELISA and microscopy. Of these 115 stool samples, 104 samples were positive by real-time PCR and were further genotyped. However, only 73 samples (70.19%) were typeable as *Giardia* TPI A of 24.66% (18/73), *Giardia* TPI B of 71.23% (52/73), and *Giardia* TPI A/B of 4.11% (3/73). Among non-typeable samples, 19 were negative by microscopy but positive by ELISA, while 12 samples were positive by microscopy and ELISA.

## 4. Discussion

Intestinal parasitic infections are a significant public health burden, especially in tropical and subtropical areas with poor sanitation and high levels of poverty [3]. Thailand is located in a tropical region, which is a highly prevalent area for parasitic infections. Although public utilities have improved in many areas of Thailand, especially in rural and remote areas, they still have problems with sanitation. Our study found that the prevalence of intestinal parasitic infections was high in school-aged and preschool children, with more than half of these children infected with at least one type of intestinal parasite. Another study conducted in the Karen hill tribe in the Northern part of Thailand also found a high prevalence of intestinal parasitic infections in about half of the school children (47.7%) [10]. The high prevalence of intestinal parasitic infections found in our and other studies might be due to the study sites being located in remote areas where the water supply and sanitation are not well developed. This hypothesis is supported by the findings of a study conducted in the suburban areas of Saraburi province, located in the Central region of Thailand. The prevalence of intestinal parasitic infections in children identified in that study was 22.1% [35], which is lower than the prevalence found in our study. Another study conducted in the rural area of Nopphitam District in Nakhon Si Thammarat province also found a low prevalence of intestinal parasitic infections in children (16%) [12]. Although the difference in prevalence between our study and this study may be due to the use of different methods to identify parasitic infections. In our study, we employed direct smear cultivation using LE media and ELISA methods, whereas the study conducted in Nakhon Si Thammarat utilized formalin-ethyl acetate sedimentation concentration that might have a lower sensitivity than our methods for detecting intestinal parasitic infections.

Our participants were school and preschool children. Children in this age range have a high risk of intestinal parasitic infections due to their increased behavioral risk, frequent outdoor exposure, and poor personal cleanliness [36]. Although the prevalence of intestinal parasitic infections was high in children, most of them were caused by non-pathogenic protozoa, while a minority of them were infected with intestinal helminths. Only 2.3% of children in our study were infected with intestinal helminths, while the study conducted among school children in the southern part of Thailand reported a slightly higher prevalence of intestinal helminth infections (6.3%) than our study [11]. These discrepancies may be attributable to variances in host genetics as well as the level of personal and community hygiene and sanitation [37,38].

The most serious effect of intestinal parasitic infections in children is the increased risk of malnutrition. Malnutrition is a significant public health problem and is the leading cause of death in children under five years of age [39]. Intestinal parasitic infections increase the risk of malnutrition in children because these infections can damage a child’s internal mucosa, leading to impaired digestion and poor absorption of nutrients [3]. A deficiency in some nutrients can affect the host’s immunological function, making the body more susceptible to infectious diseases and intestinal parasites. As part of a vicious cycle, parasitic infections can exacerbate nutrient loss, resulting in stunted growth and low nutritional status. Our study did not identify a significant association between overall intestinal parasitic infections and being underweight or stunting or wasting in children. This aligns with the majority of previous studies, which also did not reveal a significant association between intestinal parasitic infections and being underweight in children [40]. However, when each parasite was considered separately, several previous studies showed a strong correlation between *A. lumbricoides* infection and malnutrition in children [41,42,43]. In our study, the presence of *A. lumbricoides* or *E. vermicularis* was associated with being underweight, stunted, or wasted. However, these relationships did not achieve statistical significance, possibly due to the low prevalence of these helminth infections among our participants.

Our study found a significant association between *B. hominis* and being underweight in children, while *E. nana* and *I. buetschlii* were significantly associated with stunted growth. These three intestinal protozoa are non-pathogenic protozoa and rarely cause symptoms in immunocompetent persons. Thus, it is doubtful that these protozoa can induce malnutrition in children, or these associations might be influenced by other unmeasured factors, such as inadequate food intake with an undernourished diet. *B. hominis* and *E. nana* can be transmitted from human to human via the fecal-oral route, from animal to human via the water supply (eating uncooked vegetables and drinking unboiled water) and soil [44]. Thus, infected individuals often have a low socioeconomic status and inhabit unhygienic areas. Low socioeconomic status and living in unhygienic areas are significant risk factors for malnutrition in children. Thus, the relationship between non-pathogenic protozoa and malnutrition found in our study might be confounded by these factors. However, some previous evidence suggests that *B. hominis* infection could induce both acute and chronic diarrhea, which are the causes of malnutrition in immunocompetent children and adults [45,46,47]. Therefore, the pathogenicity of *B. hominis* remains inconclusive, and additional research is required on this topic. In addition, diagnosis of non-pathogenic protozoa may be important to prevent malnutrition in children. Identification of non-pathogenic parasites is also important because it highlights fecal contamination.

For the excystation of *Giardia* cysts, our rate of 16% for in vitro establishment of *Giardia* from cysts to trophozoites is slightly lower than the rates indicated in the literature, ranging from 21% to 44% [48,49,50]. In our case, the major cause of failure to produce a high rate of excystation is the inability of trophozoites to attach to the tube surface. We speculated that genetic variations could be a significant factor that contributes to the adaptability of trophozoites to survive in the culture media, as reported by Meloni and Thompson [50].

*G. lamblia* isolates from this study exhibited variation in drug susceptibility, with the greatest variability displayed in response to furazolidone, followed by metronidazole and mebendazole. Based on the in vitro drug susceptibility data, our *G. lamblia* isolates showed the least susceptibility to metronidazole among the three drugs tested. In other words, metronidazole is considered the least effective for giardiasis treatment in this study. Our finding was concordant with a previous report in Thailand that illustrated that metronidazole was less effective in giardiasis treatment in comparison to tinidazole and ornidazole [51]. To conclude that our *G. lamblia* isolates are resistant to metronidazole, mebendazole, and furazolidone, further investigations of drug susceptibility and molecular studies are needed to include *G. lamblia* susceptible strain to these three drugs as a control.

Metronidazole is the first-line treatment option for giardiasis. It is low cost, available over-the-counter, and a broad-spectrum therapeutic. Importantly, these factors contribute to its overuse which potentially drives the increasing emergence of metronidazole-resistant *Giardia* [52,53]. In most countries, the efficacy rate of metronidazole has been reported in ranges of 60–100% [54]. However, metronidazole treatment failure has been identified in a wide range between 15 and 70% after the 5–7 days course of standard treatment [13,23,55]. Furthermore, inappropriate dosing, e.g., metronidazole of <500 mg/day, could contribute to resistance in *Giardia* [18]. In recent years, numerous reports have addressed the increasing *Giardia* treatment failure of metronidazole [13,20]. A recent study demonstrated that approximately 20% of patients with chronic giardiasis had refractory to tinidazole or metronidazole, and as high as 70% of the refractory infections were originally from Asia [55]. Combination therapy of metronidazole with another drug was also recommended to improve treatment efficacy against resistant *Giardia* infections [17,56]. Further, genetic variation of key enzymes or proteins involved in the metronidazole metabolizing pathway was found to be highly associated with developing metronidazole resistance among *Giardia* clinical isolates [57,58,59]. Of note, *G. lamblia* trophozoites isolated in this study presented variable susceptibility to common antigiardial agents. It is imperative to further investigate the metabolomic profiling of these *G. lamblia* isolates. This significant data could provide a better understanding of gene variation associated with drug resistance. It is also interesting to explore the drug susceptibility of these *G. lamblia* isolates against other drugs that are frequently used for giardiasis treatment, such as tinidazole, albendazole, and nitazoxanide. Knowing the drug susceptibility pattern of *G. lamblia* isolates circulating within this high prevalence area could assist researchers in this field in extending knowledge on drug susceptibility distributions.

To the best of our knowledge, there is no report indicating metronidazole resistance in *G. lamblia* isolates in Thailand. A possible reason could be that the use of *G. lamblia* trophozoites isolated from human stool for studying drug susceptibility is challenging, e.g., bacterial contamination, low numbers of *Giardia* cysts, and no development of cysts into trophozoites. Based on our direct experience, these factors contribute to the failure to achieve a high success rate of excystation from cysts to trophozoites. Nevertheless, in vitro susceptibility data is highly valuable to guide physician decisions for the treatment of giardiasis, particularly with resistant parasites that are more difficult or impossible to treat with currently available regimens. Collectively, our drug susceptibility data highlights the importance of consistent monitoring of drug-resistant patterns of *Giardia*, especially for the communities in endemic areas. The data could inform public health settings and hygiene control measures to improve health management programs for effective treatment of *Giardia* infection in the area with a high prevalence of *Giardia* infection. Ultimately, this will reduce the transmission of resistant *Giardia* as an infectious intestinal parasite to the environment.

In this study, several methods were performed to detect *G. lamblia* to ensure that all of the *G. lamblia-positive* samples were included in the genotyping identification. *G. lamblia* assemblage B, based on TPI, was the most common assemblage identified in our study, which is similar to a report from Cambodia [60]. However, there is a considerable geographical variation of the common assemblage reported in Thailand, where assemblage A (subtype AII) was common in a province in central Thailand [61], but assemblage B was common among hill tribe children in northern Thailand [62]. The discordance between detection methods was not unprecedented, as each method detected different components of the pathogen. The samples that were not typeable could potentially be of other assemblages that were not tested for, specifically assemblage E [63].

### Strengths and Limitations

Our study has several strengths. We measured the prevalence of intestinal parasitic infections and assessed the association between these infections and chronic malnutrition, which is a significant health problem in school-aged children in an endemic area. Cultivation of intestinal parasites using LE media was applied in our study to increase the chance of parasitic detection. Moreover, drug susceptibility testing was performed for *G. lamblia*, one of the most prevalent pathogenic protozoa in Thailand [15]. Furthermore, our findings provide significant data on the variable susceptibility of *G. lamblia* to common drugs used for the treatment of giardiasis.

However, our study has some limitations. Firstly, our study design is a cross-sectional study. Therefore, no causal relationship between any association between parasitic infections and chronic malnutrition found in our study can be assumed. In addition, some factors related to both parasitic infections and malnutrition were not measured in our study. Therefore, our findings may be prone to confounding bias. Moreover, recruiting participants by invitation might affect the validity of the findings from the study because the characteristics of people who were willing or not willing to participate in the study might have different traits, such as socioeconomic status. The differences in these characteristics might be associated with the prevalence of intestinal parasitic infections and might deviate the results from the truth. Additionally, only microscopic examination was applied to detect helminthic infections. Thus, the prevalence of helminthic infections found in our study might be underestimated. In addition, there are limitations on the interpretation of drug susceptibility because the control strain(s) susceptible to the three drugs tested were not available. In this regard, we can only compare variable susceptibility among the *G. lamblia* isolates. The genotyping was limited to the identification of two assemblages of one gene based on real-time PCR due to an already wide breadth of the surveillance scope. However, the genotyping results show that assemblages A and B are the predominant assemblages, while untypeable samples can be subject to further characterization.

## 5. Conclusions

Our findings highlight the ongoing burden of intestinal parasite infections in Thai children living in remote places, as well as the potential emergence of anti-parasitic resistance in *G. lamblia*. In addition, infections with *G. lamblia*, *B. hominis*, and *E. nana* were associated with children being underweight. As a result, early screening and diagnosis of intestinal parasitic infections are important for eliminating these intestinal parasite diseases and preventing malnutrition in children.

## Figures and Tables

**Figure 1 tropicalmed-08-00394-f001:**
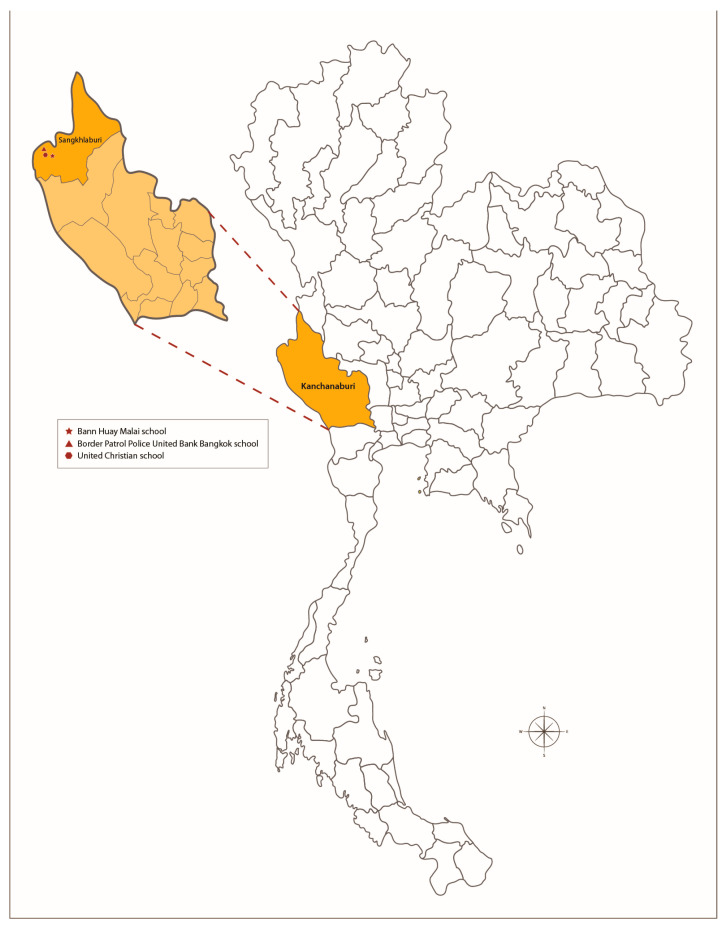
Location of the study sites at Bann Huay Malai, Border Patrol Police United Bank Bangkok, and United Christian schools, Sangkhlaburi District, Kanchanaburi Province, Thailand.

**Figure 2 tropicalmed-08-00394-f002:**
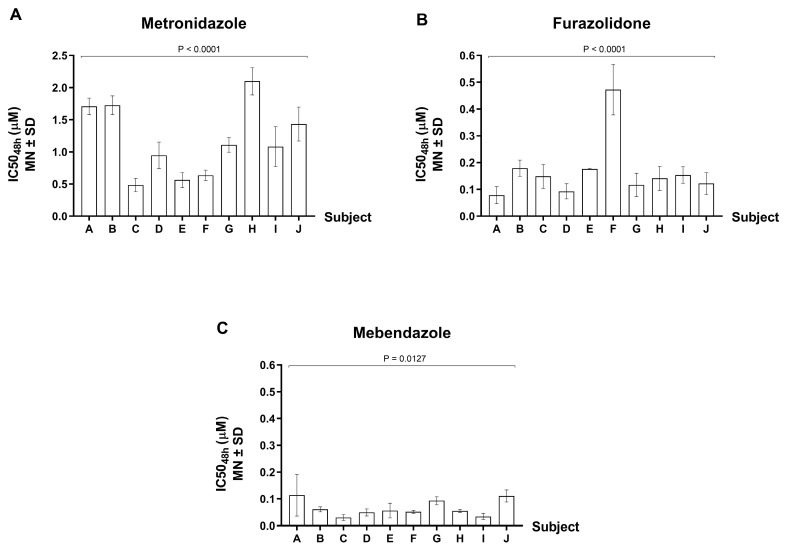
Drug susceptibility test of *G. lamblia* isolates from 10 volunteers. The data are expressed as IC_50_ (µM) ± SD of three independent experiments. X-axis describes individual subjects: (**A**) Metronidazole; (**B**) Furazolidone; (**C**) Mebendazole. The subjects designated as A–J in the X-axis were de-identified from specimen ID without linking to the identifying subject information.

**Table 1 tropicalmed-08-00394-t001:** Characteristics of study participants.

Characteristic	Total Participant N = 661	Parasitic Infection N = 445 (67.32%)	Non-Parasitic Infection N = 216 (32.68%)	Chi^2^ Value, *p*-Value
Age; mean (SD); year	8.29 (2.14)	8.36 (2.11)	8.12 (2.22)	0.174
Sex				
- male	304 (45.99)	198 (44.49)	106 (49.07)	1.228, 0.268
- female	357 (54.01)	247 (55.51)	110 (50.93)	
Height; mean (SD); cm	126.54 (13.01)	126.29 (12.84)	127.07 (13.35)	0.469
Weight; mean (SD); kg	26.72 (9.56)	26.35 (9.34)	27.47 (9.97)	0.160
Stool characteristics				
- hard	216 (32.68)	137 (30.79)	79 (36.57)	2.214, 0.137
- soft	392 (59.30)	273 (61.35)	119 (59.05)	2.358, 0.125
- loose	49 (7.41)	32 (7.19)	17 (7.87)	0.098, 0.755
- watery	4 (0.61)	3 (0.67)	1 (0.46)	0.108, 1.00
- rice watery	0 (0)	0 (0)	0 (0)	1.00
Symptoms & signs				
- Abdominal cramping	64 (9.68)	49 (11.01)	15 (6.94)	2.750, 0.097
- Nausea	10 (1.51)	8 (1.80)	2 (0.93)	0.742, 0.511
- Vomiting	30 (4.54)	22 (4.94)	8 (3.70)	0.516, 0.472
- Diarrhea	27 (4.08)	22 (4.94)	5 (2.31)	2.565, 0.109
- Fever	36 (5.45)	27 (6.07)	9 (4.17)	1.020, 0.312
- RBCs in stool	0 (0)	0 (0)	0 (0)	1.00
- WBCs in stool	9 (1.36)	8 (1.79)	1 (0.46)	1.929, 0.367
Study’s site				
- Bann Huay Malai School	226 (34.19)	152 (67.26)	74 (32.74)	10.70, 0.005
- Border Patrol Police United Bank Bangkok School	110 (16.64)	88 (80.0)	22 (20.0)	
- United Christian School	325 (49.17)	205 (63.08)	120 (36.92)	

**Table 2 tropicalmed-08-00394-t002:** Prevalence of intestinal parasitic infections.

Organisms	Prevalence (%) (95% Confidence Interval)		
	Total	Preschool Age *	School Age *
**Overall infection**			
- Helminths	0.30 (0.04–1.09)	1.10 (0.03–5.97)	0.18 (0.004–0.97)
- Protozoa	65.05 (61.28–68.69)	56.04 (45.25–66.44)	66.49 (62.45–70.36)
- Both helminth and protozoa	1.97 (1.05–3.34)	3.30 (0.69–9.33)	1.75 (0.84–3.20)
**Protozoa infection**			
**Non-pathogenic**			
- *Blastocystis hominis*	49.32 (45.44–53.22)	36.26 (26.44–47.01)	51.40 (47.21–55.58)
- *Endolimax nana*	26.93 (23.58–30.48)	24.18 (15.81–34.28)	27.37 (23.75–31.23)
- *Entamoeba coli*	9.53 (7.40–12.03)	3.30 (0.69–9.33)	10.53 (8.13–13.34)
- *Iodamoeba buetschlii*	5.45 (3.84–7.46)	3.30 (0.69–9.33)	5.79 (4.02–8.03)
- *Entamoeba* spp.	3.03 (1.86–4.63)	1.10 (0.03–5.97)	3.33 (2.02–5.16)
- *Pentatrichomonas hominis*	1.97 (1.05–3.34)	1.10 (0.03–5.97)	2.11 (1.09–3.65)
**Pathogenic**			
- *Entamoeba histolytica*	0.76 (0.25–1.76)	0 (0–3.97)	0.88 (0.29–2.04)
- *Giardia lamblia*	17.40 (14.58–20.51)	28.57 (19.59–39.0)	15.61 (12.73–18.86)
- *Cryptosporidium*	0 (0–0.56)	0 (0–3.97)	0 (0–0.65)
**Helminth infection**			
- *Ascaris lumbricoides*	0.91 (0.33–1.97)	2.20 (0.27–7.71)	0.70 (0.19–1.79)
- *Hookworm*	0.45 (0.09–1.32)	0 (0–3.97)	0.53 (0.11–1.53)
- *Enterobius vermicularis*	0.61 (0.17–1.54)	2.20 (0.27–7.71)	0.35 (0.04–1.26)
- *Strongyloides stercoralis*	0.15 (0.004–0.84)	0 (0–3.97)	0.18 (0.004–0.97)
- *Trichuris trichiura*	0.15 (0.004–0.84)	0 (0–3.97)	0.18 (0.004–0.97)

* Preschool age is defined as age 3 to 5 years, and school-age is defined as age 6 to 12 years.

**Table 3 tropicalmed-08-00394-t003:** Associations between intestinal parasitic infections and nutritional statuses.

Organism	WFA				HFA				WFH			
	Yes	No	OR (95% CI)	*p*-Value	Yes	No	OR (95% CI)	*p*-Value	Yes	No	OR (95% CI)	*p*-Value
**Overall infection**												
- Absence	21 (9.72)	195 (90.28)	1	0.070	23 (10.65)	193 (89.35)	1	0.141	16 (7.41)	200 (92.59)	1	0.915
- Presence	66 (14.83)	379 (85.17)	1.62 (0.96–2.72)		66 (14.83)	379 (85.17)	1.46 (0.88–2.42)		34 (7.64)	411 (92.36)	1.03 (0.56–1.92)	
Helminth infection												
- Absence	84 (13.0)	562 (87.0)	1	0.433	86 (13.31)	560 (86.69)	1	0.457	48 (7.43)	598 (92.57)	1	0.401
- Presence	3 (20.0)	12 (80.0)	1.67 (0.46–6.05)		3 (20.0)	12 (80.0)	1.63 (0.45–5.89)		2 (13.33)	13 (86.67)	1.92 (0.42–8.74)	
Protozoa infection												
- Absence	21 (9.63)	197 (90.37)	1	0.062	23 (10.55)	195 (89.45)	1	0.125	16 (7.34)	202 (92.66)	1	0.878
- Presence	66 (14.90)	377 (85.10)	1.64 (0.98–2.76)		66 (14.90)	377 (85.10)	1.48 (0.90–2.46)		34 (7.67)	409 (92.33)	1.05 (0.57–1.95)	
**Protozoa infection**												
*B. hominis*												
- Absence	34 (10.15)	301 (89.85)	1	0.021	37 (11.04)	298 (88.96)	1	0.066	25 (7.46)	310 (92.54)	1	0.920
- Presence	53 (16.26)	273 (83.74)	1.72 (1.08–2.72)		52 (15.95)	274 (84.05)	1.53 (0.97–2.40)		25 (7.67)	301 (92.33)	1.03 (0.58–1.83)	
*E. coli*												
- Absence	81 (13.55)	517 (86.45)	1	0.372	83 (13.88)	515 (86.12)	1	0.339	46 (7.69)	552 (92.31)	1	0.702
- Presence	6 (9.52)	57 (90.48)	0.67 (0.28–1.61)		6 (9.52)	57 (90.48)	0.65 (0.27–1.56)		4 (6.35)	59 (93.65)	0.81 (0.28–2.34)	
*E. nana*												
- Absence	57 (11.80)	428 (88.2)	1	0.090	57 (11.80)	426 (88.20)	1	0.040	37 (7.66)	446 (92.34)	1	0.878
- Presence	30 (16.85)	148 (83.15)	1.51 (0.94–2.45)		32 (17.98)	146 (82.02)	1.64 (1.02–2.63)		13 (7.30)	165 (92.70)	0.95 (0.49–1.83)	
*G. lamblia*												
- Absence	73 (13.37)	473 (86.63)	1	0.730	77 (14.10)	469 (85.90)	1	0.297	44 (8.06)	502 (91.94)	1	0.299
- Presence	14 (12.17)	101 (87.83)	0.90 (0.49–1.65)		12 (10.43)	103 (89.57)	0.71 (0.37–1.35)		6 (5.22)	109 (94.78)	0.63 (0.26–1.51)	
*P. hominis*												
- Absence	85 (13.12)	563 (86.88)	1	0.811	86 (13.27)	562 (86.73)	1	0.314	49 (7.56)	599 (92.44)	1	0.986
- Presence	2 (15.38)	11 (84.62)	1.20 (0.26–5.53)		3 (23.08)	10 (76.92)	1.96 (0.53–7.27)		1 (7.69)	12 (92.31)	1.02 (0.13–8.00)	
*I. buetschlii*												
- Absence	79 (12.64)	546 (87.36)	1	0.104	80 (12.80)	545 (87.2)	1	0.042	47 (7.52)	578 (92.48)	1	0.858
- Presence	8 (22.22)	28 (27.78)	1.97 (0.87–4.49)		9 (25.0)	27 (75.0)	2.27 (1.03–5.00)		3 (8.33)	33 (91.67)	1.12 (0.33–3.78)	
Most prevalent protozoa (*G. lamblia, B. Hominis, and E. nana*)												
- Absence	22 (8.94)	224 (91.06)	1	0.015	25 (10.16)	221 (89.84)	1	0.057	18 (7.32)	228 (92.68)	1	0.853
- Presence	350 (84.34)	65 (15.66)	1.89 (1.13–3.15)		64 (15.42)	351 (84.58)	1.61 (0.99–2.64)		32 (7.71)	383 (92.29)	1.06 (0.58–1.93)	
**Helminth infection**												
*A. lumbricoides*												
- Absence	86 (13.13)	569 (86.87)	1	0.799	87 (13.28)	568 (86.72)	1	0.176	49 (7.48)	606 (92.52)	1	0.413
- Presence	1 (16.67)	5 (83.33)	1.32 (0.15–11.46)		2 (33.33)	4 (66.67)	3.26 (0.59–18.09)		1 (16.67)	5 (83.33)	2.47 (0.28–21.59)	
*Hookworm*												
- Absence	87 (13.22)	571 (86.78)	-	-	89 (13.53)	569 (86.47)	-	-	50 (7.6)	608 (92.4)	-	-
- Presence	0 (0.0)	3 (100)	-		0 (0.0)	3 (100)	-		0 (0.0)	3 (100)	-	
*E. vermicularis*												
- Absence	85 (12.94)	572 (87.06)	1	0.058	88 (13.39)	569 (86.61)	1	0.508	49 (7.46)	608 (92.54)	1	0.223
- Presence	2 (50.0)	2 (50.0)	6.73 (0.94–48.41)		1 (25.0)	3 (75.0)	2.16 (0.22–20.95)		1 (25.0)	3 (75.0)	4.14 (0.42–40.51)	
*S. stercoralis*												
- Absence	87 (13.18)	573 (86.82)	-	-	89 (13.48)	571 (86.52)	-	-	50 (7.58)	610 (92.42)	-	-
- Presence	0 (0.0)	1 (100)	-		0 (0.0)	1 (100)	-		0 (0.0)	1 (100)	-	
*T. trichiura*												
- Absence	87 (13.18)	573 (86.82)	-	-	89 (13.48)	571 (86.52)	-	-	50 (7.58)	610 (92.42)	-	-
- Presence	0 (0.0)	1 (100)	-		0 (0.0)	1 (100)	-		0 (0.0)	1 (100)	-	

CI, confidence interval; OR, odds ratio; WFA, weight for Age; HFA, height for age; WFH, weight for height.

## Data Availability

Data is available upon request after approval by the research committee.

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
