# Peer review of "Prevalence of Intestinal Parasitic Infections, Genotypes, and Drug Susceptibility of Giardia lamblia among Preschool and School-Aged Children: A Cross-Sectional Study in Thailand"

_tropicalmed, 2023, doi:10.3390/tropicalmed8080394_

Round 1

Reviewer 1 Report

The manuscript “Prevalence of intestinal parasitic infections, genotypes and drug susceptibility of Giardia lamblia among preschool and school aged children: A cross-sectional study in Thailand” presents data from: 1) a cross-sectional study of parasitological investigation of feces in three schools in Sangkhlaburi District, Kanchanaburi Province, 2) anthropometric measurements of children included in the study; 3) three-drug G. lamblia susceptibility assay on some isolates from clinical samples; 4) G. lamblia genotyping.

The study presents relevant data on the frequency of intestinal protozoa (high) and helminths (low) in the region. The manuscript is clearly written and easy to understand. However, the experimental design of the study does not seem well defined. It is difficult for the reader to understand the rationale of the study, as the central question is not clear and the introduction does not provide subsidies for this understanding.

Therefore, some considerations are necessary:

 1) In the introduction, the themes are not addressed: genotyping of G. lamblia and susceptibility to drugs. The lack of these subjects makes it difficult to understand why they are part of this study. How important are they in this context?

 2) In Materials and Methods, in “An aliquot of each stool specimen (approximately 0.5-2 mL) was identified for E. histolytica, G. lamblia, and Cryptosporidium using commercial ELISA kits (TechLab, Inc., USA). Procedure was performed according to the manufacturer’s instructions.” Put “Cryptosporidium” in italics.

 3) In Results, in “…any significant difference, except that the participants from the Border 178 Patrol Police United Bank Bangkok School had a higher rate of intestinal parasitic infec- 179 tions (80%) than the participants from Bann Huay Malai 67.26%) and United Christian 180 School (63.08%).” The "(" is missing before the percentage value.

 4) “Four hundred and forty-five students had intestinal parasitic infection with the prevalence of 67.32% (95% CI: 63.60%-70.89%). Protozoa, helminth, and both protozoa and helminth infections were present in 430, 2, and 13 of the students, with prevalence of 65.05 (95% CI: 61.28%-68.69%), 0.3% (95% CI: 0.04%-1.09%), and 1.97% (95% CI: 1.05%-3.34%), respectively.” it is important to differentiate the percentages per school studied.

 5)    In “3.3 Excystation of G. lambia cysts and drug susceptibility test.”, “Of the 115 stools that were positive for G. lamblia by either or both ELISA and microscopy, 85 stools were selected for cyst purification.” What were the criteria for selecting the 85/115 stool samples for cyst purification?

 6) “For metronidazole, the IC50 values varied from 0.484 to 2.1 μM, representing a 4.3-fold variation of susceptibility. For furazolidone, the IC50 values varied from 0.079 to 0.473 μM, representing a 6-fold variation of susceptibility. For mebendazole, the IC50 values varied from 0.030 to 0.114 μM, representing a 3.8-fold variation of susceptibility. In addition, our results illustrated that the mean IC50 values of 5 Giardia isolates (5/10; 50%) were slightly greater than the mean values of the WB Clone C6 reference isolate; 1 isolate had 1.2-fold increase for metronidazole, 1 isolate had 1.37-fold increase for furazolidone, and 3 isolates; had a range of 1.44 -1.76-fold increase for mebendazole.”  Add that the IC50 increases observed for the three drugs used are numerical only and do not have statistical power. In fact, one of the 3 isolates (47) that showed less susceptibility to mebendazole, showed a very large standard deviation. This deviation could attribute similar IC50 values to the WB strain to the isolate.

7) In “G. lamblia genotypes”, in “Among non-typable samples, 19 were negative by microscopy but positive by ELISA” Among the samples that were negative in the microscopy and positive in the ELISA, would it not be possible to be a false positive of the immunological diagnosis? This can be better discussed.

 8) In “Discussion”, “Among the protozoa infection, nonpathogenic protozoa were more prevalent than pathogenic protozoa with Blastocystis hominis, and G. lamblia being the most prevalent nonpathogenic and pathogenic protozoa in our subjects, respectively” Rephrase the sentence. Very repetitive.

 9) In “These discrepancies may be attributable to variances in host genetics and geographical circumstances, as well as the level of personal and community hygiene and sanitation” Faltou mencionar contaminação do solo que justificaria os geohelmintos.

 10) In “In our study, the presence of Ascaris lumbricoides or Enterobius vermicularis was also found to be strongly associated with being underweight, stunning, or wasting. However, these relationships did not achieve statistical significance, possibly due to the low prevalence of these helminth infections among our participants.” Strongly remove. If there was no statistical significance, it could not be considered a strong association.

 11) In “Of the 3 drugs tested” replace “3” to “three”.

 12) In table 2 replace “Blastocystis Hominis” to “Blastocystis hominis”.

General considerations:

1-    In the same way that the data on genotyping and susceptibility to drugs was not presented in the introduction, the topics again do not appear in the discussion.

Regarding genotyping, what is the importance of identifying these assemblages? What do other studies reported about the assemblages circulating in the region? It is necessary to point out the limitation of the study in not carrying out the sequencing of the isolates, because other assemblages could be identified, such as assemblage E, for example.

Regarding the susceptibility tests, it is important to consider that it has been previously described that the WB C6 strain ATCC50803 has some resistance to metronidazole. Strain WB was originally isolated from a patient with refractory giardiasis (Smith et al. 1982). And therefore, the WB strain should only be used as a control in susceptibility tests when compared with itself. A strain susceptible to the 3 drugs should have been used as a control.

I suggest removing the susceptibility experiments from the manuscript. A low percentage of isolates successfully excised and the WB strain used as a control is not recommended for this evaluation.

2-    The studied children were grouped in the same group. It is important to separate by age group. In the range of 3 to 12 years, children are very different: in habits, microbiota, immune response stimulation, for example. At a minimum, children should be separated into preschoolers and schoolchildren and the data reanalyzed.  

3-    In the manuscript it is mentioned that most of the children were asymptomatic. Although G. lamblia infection can cause physical development delay in both symptomatic and asymptomatic children, the absence of symptoms could justify the study's anthropometric data.

In addition, anthropometric parameters vary greatly between populations of different nationalities. The comparison of anthropometric data can be redone comparing only individuals negative for all parasites and positive for the most prevalent protozoa (G. lamblia, B. hominis and E. nana).

4-    Symptomatology data were little explored. It would be interesting to associate 1) symptom with infective enteroparasite; 2) Symptom and G. lamblia assemblage identified.

5-     The manuscript presents data on the high frequency of protozoa considered non-pathogenic. In the discussion, the importance of diagnosing non-pathogenic protozoa should be mentioned.

6-    The conclusion must approach the main findings responding to the proposed objective. Therefore, the conclusion needs to be rewritten. The current wording points to developments indicated from the results of this research.

Author Response

Response to Reviewer 1’s comments

The manuscript “Prevalence of intestinal parasitic infections, genotypes and drug susceptibility of Giardia lamblia among preschool and school aged children: A cross-sectional study in Thailand” presents data from: 1) a cross-sectional study of parasitological investigation of feces in three schools in Sangkhlaburi District, Kanchanaburi Province, 2) anthropometric measurements of children included in the study; 3) three-drug G. lamblia susceptibility assay on some isolates from clinical samples; 4) G. lamblia genotyping.

The study presents relevant data on the frequency of intestinal protozoa (high) and helminths (low) in the region. The manuscript is clearly written and easy to understand. However, the experimental design of the study does not seem well defined. It is difficult for the reader to understand the rationale of the study, as the central question is not clear and the introduction does not provide subsidies for this understanding.

Therefore, some considerations are necessary:

 1) In the introduction, the themes are not addressed: genotyping of G. lamblia and susceptibility to drugs. The lack of these subjects makes it difficult to understand why they are part of this study. How important are they in this context?

Response: Background and rationale for genotyping and drug susceptibility test for G. lamblia were included in the introduction part (please see lines 60-75).

 2) In Materials and Methods, in “An aliquot of each stool specimen (approximately 0.5-2 mL) was identified for E. histolytica, G. lamblia, and Cryptosporidium using commercial ELISA kits (TechLab, Inc., USA). Procedure was performed according to the manufacturer’s instructions.” Put “Cryptosporidium” in italics.

Response: Corrected.

 3) In Results, in “…any significant difference, except that the participants from the Border 178 Patrol Police United Bank Bangkok School had a higher rate of intestinal parasitic infec- 179 tions (80%) than the participants from Bann Huay Malai 67.26%) and United Christian 180 School (63.08%).” The "(" is missing before the percentage value.

Response: Corrected.

 4) “Four hundred and forty-five students had intestinal parasitic infection with the prevalence of 67.32% (95% CI: 63.60%-70.89%). Protozoa, helminth, and both protozoa and helminth infections were present in 430, 2, and 13 of the students, with prevalence of 65.05 (95% CI: 61.28%-68.69%), 0.3% (95% CI: 0.04%-1.09%), and 1.97% (95% CI: 1.05%-3.34%), respectively.” it is important to differentiate the percentages per school studied.

Response: Prevalence of parasitic infections for each school have been added in the results (please see lines 264-266).

 5)    In “3.3 Excystation of G. lambia cysts and drug susceptibility test.”, “Of the 115 stools that were positive for G. lamblia by either or both ELISA and microscopy, 85 stools were selected for cyst purification.” What were the criteria for selecting the 85/115 stool samples for cyst purification?

Response: The criteria used for selection stool samples for cyst purification has been explained (please see lines 317-318).

 6) “For metronidazole, the IC50 values varied from 0.484 to 2.1 μM, representing a 4.3-fold variation of susceptibility. For furazolidone, the IC50 values varied from 0.079 to 0.473 μM, representing a 6-fold variation of susceptibility. For mebendazole, the IC50 values varied from 0.030 to 0.114 μM, representing a 3.8-fold variation of susceptibility. In addition, our results illustrated that the mean IC50 values of 5 Giardia isolates (5/10; 50%) were slightly greater than the mean values of the WB Clone C6 reference isolate; 1 isolate had 1.2-fold increase for metronidazole, 1 isolate had 1.37-fold increase for furazolidone, and 3 isolates; had a range of 1.44 -1.76-fold increase for mebendazole.”  Add that the IC50 increases observed for the three drugs used are numerical only and do not have statistical power. In fact, one of the 3 isolates (47) that showed less susceptibility to mebendazole, showed a very large standard deviation. This deviation could attribute similar IC50 values to the WB strain to the isolate.

Response: Since WB Clone C6 isolate is resistant to some drug but not all 3 drugs tested. We reconsider to remove the WB Clone C6 isolate from drug susceptibility part to prevent misinterpretation of drug susceptibility. The data was reanalysed to compare IC50 across the 10 Giardia isolates for each drug tested (please see lines 332-354, Figure 2, and Figure S1 and S2).

7) In “G. lamblia genotypes”, in “Among non-typable samples, 19 were negative by microscopy but positive by ELISA” Among the samples that were negative in the microscopy and positive in the ELISA, would it not be possible to be a false positive of the immunological diagnosis? This can be better discussed.

Response: This concern was described in the discussion part (please see lines 396-400).

 8) In “Discussion”, “Among the protozoa infection, nonpathogenic protozoa were more prevalent than pathogenic protozoa with Blastocystis hominis, and G. lamblia being the most prevalent nonpathogenic and pathogenic protozoa in our subjects, respectively” Rephrase the sentence. Very repetitive.

Response: We have revised this sentence as suggested (please see lines 385-389).

 9) In “These discrepancies may be attributable to variances in host genetics and geographical circumstances, as well as the level of personal and community hygiene and sanitation” Faltou mencionar contaminação do solo que justificaria os geohelmintos.

Response: We have deleted the term “geographical circumstances” from this sentence (please see line 432-433).

 10) In “In our study, the presence of Ascaris lumbricoides or Enterobius vermicularis was also found to be strongly associated with being underweight, stunning, or wasting. However, these relationships did not achieve statistical significance, possibly due to the low prevalence of these helminth infections among our participants.” Strongly remove. If there was no statistical significance, it could not be considered a strong association.

Response: We have deleted the word “Strongly” from the sentence (please see lines 447-450).

 11) In “Of the 3 drugs tested” replace “3” to “three”.

Response: This sentence has been deleted due to the comments from other reviewers.

 12) In table 2 replace “Blastocystis Hominis” to “Blastocystis hominis”.

Response: Corrected.

 General considerations:

  1. In the same way that the data on genotyping and susceptibility to drugs was not presented in the introduction, the topics again do not appear in the discussion.

Response: The rational for conducting the genotyping and drug susceptibility of Giardia has been stated in the introduction part (lines 60-74) and the data has been discussed in the discussion part (please see lines 390-400 and 476-524).

  1. Regarding genotyping, what is the importance of identifying these assemblages? What do other studies reported about the assemblages circulating in the region? It is necessary to point out the limitation of the study in not carrying out the sequencing of the isolates, because other assemblages could be identified, such as assemblage E, for example.

Response: This information was described in the discussion part (please see lines 390-400).

  1. Regarding the susceptibility tests, it is important to consider that it has been previously described that the WB C6 strain ATCC50803 has some resistance to metronidazole. Strain WB was originally isolated from a patient with refractory giardiasis (Smith et al. 1982). And therefore, the WB strain should only be used as a control in susceptibility tests when compared with itself. A strain susceptible to the 3 drugs should have been used as a control.

Response: Since WB Clone C6 isolate is resistant to some drug but not all 3 drugs tested. We reconsider to remove the WB Clone C6 isolate from drug susceptibility part to prevent misinterpretation of drug susceptibility. The data was reanalysed to compare IC50 across the 10 Giardia isolates for each drug tested (please see lines 332-354).

  1. I suggest removing the susceptibility experiments from the manuscript. A low percentage of isolates successfully excised and the WB strain used as a control is not recommended for this evaluation.

Response: We acknowledged your comment and removed the WB strain from evaluation of drug susceptibility. However, we prefer to remain the drug susceptibility experiment in this manuscript. Although we obtained the low success rate of cyst excystation into trophozoites however the results of drug susceptibility from this study would be helpful. We consider that our data could provide the comprehensive knowledge of distributing drug susceptibility of G. lambia isolates circulating within this high prevalence area of Giardia infection. This could assist researchers in this filed to extend knowledge on drug susceptibility distributions.

  1. The studied children were grouped in the same group. It is important to separate by age group. In the range of 3 to 12 years, children are very different: in habits, microbiota, immune response stimulation, for example. At a minimum, children should be separated into preschoolers and schoolchildren and the data reanalyzed.

Response: Subgroup analysis according to age group has been performed and the findings have been added in the results part (please see lines 273-276, 283-285, 289-290, and Table 2).

  1. In the manuscript it is mentioned that most of the children were asymptomatic. Although lamblia infection can cause physical development delay in both symptomatic and asymptomatic children, the absence of symptoms could justify the study's anthropometric data.

Response: Thank you very much for your comments.

  1. In addition, anthropometric parameters vary greatly between populations of different nationalities. The comparison of anthropometric data can be redone comparing only individuals negative for all parasites and positive for the most prevalent protozoa ( lamblia, B. hominis and E. nana).

Response: The association between the most prevalent protozoa and anthropometric parameters has been performed and the findings are added in the results (please see lines 301-304 and Table 3).

  1. Symptomatology data were little explored. It would be interesting to associate 1) symptom with infective enteroparasite; 2) Symptom and lamblia assemblage identified.

Response: We have assessed the difference of symptoms and signs between children infected and not-infected with pathogenic parasites (i.e., G. lamblia, Entamoeba histolyt-ica and intestinal helminths). The findings are added in the results (please see lines 256-259 and Table S2).

  1. The manuscript presents data on the high frequency of protozoa considered non-pathogenic. In the discussion, the importance of diagnosing non-pathogenic protozoa should be mentioned.

Response: We have discussed this point in the discussion part (please see lines 467-469).

  1. The conclusion must approach the main findings responding to the proposed objective. Therefore, the conclusion needs to be rewritten. The current wording points to developments indicated from the results of this research.

Response: We have revised the conclusion according to your comment (lines 556-558).

Reviewer 2 Report

In the introduction, the authors fail to explain the background and objectives of the study.

Lines 32, 116, 118, 131, and 145: The species name should be in italics.

Line 68: Add a full stop at the end of the sentence.

Lines 65–78: Add a suitable heading to these lines, like 2.1. Study area and study population. Also, update the other heading numbers accordingly.

Line 121: "ml" should be "mL." Check and correct it throughout the text.

Line 162: Clearly mention all subjects in which you used the t-test and chi-square test. Also, add chi-square values to the results.

Line 180: 67.26%) should be (67.26%)

Table 1, column 2, line 2. N = 611 or N = 661?

Table 2: What was the statistical association with parasitic infection?

The results of microscopy and ELISA were not mentioned clearly.

Which kind of parasite is prevalent in which school is not mentioned clearly.

What kind of statistical analyses were used to determine the drug's susceptibility?

What is the number of true positives (positive by microscopy and ELISA)? First of all, discuss the genotype of true positives, then positive samples, either by only microscopy or by only ELISA.

Organize the discussion to address each of the experiments or studies for which the author presented results.

Do not waste entire sentences restating your results.

Authors should necessarily make reference to the findings of others in order to support their interpretations.

In the introduction, the authors fail to explain the background and objectives of the study.

Lines 32, 116, 118, 131, and 145: The species name should be in italics.

Line 68: Add a full stop at the end of the sentence.

Lines 65–78: Add a suitable heading to these lines, like 2.1. Study area and study population. Also, update the other heading numbers accordingly.

Line 121: "ml" should be "mL." Check and correct it throughout the text.

Line 162: Clearly mention all subjects in which you used the t-test and chi-square test. Also, add chi-square values to the results.

Line 180: 67.26%) should be (67.26%)

Table 1, column 2, line 2. N = 611 or N = 661?

Table 2: What was the statistical association with parasitic infection?

The results of microscopy and ELISA were not mentioned clearly.

Which kind of parasite is prevalent in which school is not mentioned clearly.

What kind of statistical analyses were used to determine the drug's susceptibility?

What is the number of true positives (positive by microscopy and ELISA)? First of all, discuss the genotype of true positives, then positive samples, either by only microscopy or by only ELISA.

Organize the discussion to address each of the experiments or studies for which the author presented results.

Do not waste entire sentences restating your results.

Authors should necessarily make reference to the findings of others in order to support their interpretations.

Author Response

Response to Reviewer 2’s comments

  1. In the introduction, the authors fail to explain the background and objectives of the study.

Response: Background and objectives for genotyping and drug susceptibility test for G. lamblia were included in the introduction part (please see lines 60-79).

  1. Lines 32, 116, 118, 131, and 145: The species name should be in italics.

Response: These species name has been revised to be in italics.

  1. Line 68: Add a full stop at the end of the sentence.

Response: Added.

  1. Lines 65–78: Add a suitable heading to these lines, like 2.1. Study area and study population. Also, update the other heading numbers accordingly.

Response: Added and updated the heading numbers accordingly.

  1. Line 121: "ml" should be "mL." Check and correct it throughout the text.

Response: Corrected.

  1. Line 162: Clearly mention all subjects in which you used the t-test and chi-square test. Also, add chi-square values to the results.

Response: Types of data for t-test and chi-square test have been added in the statistical analysis part (please see line 234). Chi-square values have been added in Table 1.

  1. Line 180: 67.26%) should be (67.26%)

Response: Added.

  1. Table 1, column 2, line 2. N = 611 or N = 661?

Response: Corrected.

  1. Table 2: What was the statistical association with parasitic infection?

Response: Table 2 presents only prevalence and their 95% confidence interval of parasitic infection in children. The association between parasitic infection and nutritional status is presented in Table 3.

  1. The results of microscopy and ELISA were not mentioned clearly.

Response: The results of microscopy and ELISA have been stated in the results (Section 3.5, lines 374-376).

  1. Which kind of parasite is prevalent in which school is not mentioned clearly.

Response: Prevalence of parasitic infection according to the study sites has been added in the results (please see Section 3.1, lines 264-265, and 269-276).

  1. What kind of statistical analyses were used to determine the drug's susceptibility?

Response: Method used for statistical analysis of drug susceptibility was described in the statistical analysis section (please see lines 236-238).

  1. What is the number of true positives (positive by microscopy and ELISA)? First of all, discuss the genotype of true positives, then positive samples, either by only microscopy or by only ELISA.

Response: The results of microscopy and ELISA have been added (Section 3.5, lines 374-376).

  1. Organize the discussion to address each of the experiments or studies for which the author presented results.

Response. We have re-organized the discussion part.

  1. Do not waste entire sentences restating your results.

Response: We have revised and edited the English language for the whole manuscript.

  1. Authors should necessarily make reference to the findings of others in order to support their interpretations.

Response: The references of other studies have been cited and discussed in the discussion part.

Reviewer 3 Report

In the present manuscript, the authors estimated the prevalence of intestinal parasitic infections and evaluated the association between intestinal parasitic infections and nutritional status in children living in a district of a province in Thailand. In addition, genotypes and drug susceptibility of Giardia lamblia were also investigated.

Some comments:

Authors should expand the introduction section by adding information on drug resistance and genotypes/assemblages of G. lamblia.

Once the species are named, use the abbreviated genus in subsequent mentions.

Why “Giardia (G.) lamblia” and not “Giardia lamblia”? (lines 18 and 54).

A small map with the geographic location of the sites and the province of the study could be helpful.

Could the fact that the participants were recruited by invitation have affected the prevalence found?

Line 116: “and Cryptosporidium by ELISA”; however, nothing is mentioned about it in the manuscript.

For the 2.4.5 section, more details should be included.

Table 1: please correct the number of total participants.

Is there any specific reason that could explain the significant higher rate of intestinal parasitic infections found in the children from the Border Patrol Police United Bank Bangkok School?

Line 195: please change “9.53%” to “3.03%”.

The prevalence of Enterobious vermicularis was 0.61%; however, this prevalence could have been much higher if a Graham Test had been performed, right?

Lines 209-215: please indicate the p-values. It would also be appropriate to indicate in some way the statically significant cases in Table 3.

A discussion of the results obtained in relation to genotyping is missing.

Lines 374-375: “the majority of our Giardia isolates seemed to be resistant to mebendazole as IC50 values in 30% isolates were higher than the reference isolate”. 30% is not the majority; on the other hand, are there other possible reasons that could explain these results?

Line 383: “which may contribute to the development of drug resistance”.  Please indicate references related to resistance to mebendazole/benzimidazoles.

Line 387: “In other words, the Giardia isolates demonstrated an increased resistance to metronidazole”. This reviewer thinks that the fact that the IC50 values of metronidazole were relatively high and that it had less antiparasitic effect than the other drugs does not mean necessarily that the isolates studied show an increased resistance. Are there other possible reasons that could explain these results?

Authors should include a paragraph with future directions.

Articles that could be consulted and cited by the authors:

Assavapongpaiboon B, Bunkasem U, Sanprasert V, Nuchprayoon S. A Cross-Sectional Study on Intestinal Parasitic Infections in Children in Suburban Public Primary Schools, Saraburi, the Central Region of Thailand. Am J Trop Med Hyg. 2018 Mar;98(3):763-767. doi: 10.4269/ajtmh.17-0240. Epub 2018 Jan 18. PMID: 29363443; PMCID: PMC5930881.

Punsawad C, Phasuk N, Bunratsami S, Thongtup K, Viriyavejakul P, Palipoch S, Koomhin P, Nongnaul S. Prevalence of intestinal parasitic infections and associated risk factors for hookworm infections among primary schoolchildren in rural areas of Nakhon Si Thammarat, southern Thailand. BMC Public Health. 2018 Sep 14;18(1):1118. doi: 10.1186/s12889-018-6023-3. PMID: 30217180; PMCID: PMC6137929.

Minor editing of English language required

Author Response

Response to Reviewer 3’s comments

In the present manuscript, the authors estimated the prevalence of intestinal parasitic infections and evaluated the association between intestinal parasitic infections and nutritional status in children living in a district of a province in Thailand. In addition, genotypes and drug susceptibility of Giardia lamblia were also investigated.

Some comments:

  1. Authors should expand the introduction section by adding information on drug resistance and genotypes/assemblages of lamblia.

Response: Information on drug resistance and genotypes/assemblages of G. lamblia has been added in the introduction part (lines 60-74).

  1. Once the species are named, use the abbreviated genus in subsequent mentions.

Response: Revised.

  1. Why “Giardia (G.) lamblia” and not “Giardia lamblia”? (lines 18 and 54).

Response: Corrected.

  1. A small map with the geographic location of the sites and the province of the study could be helpful.

Response: The map showing the geographic location of study sites and the province of the study has been added (please see Figure 1).

  1. Could the fact that the participants were recruited by invitation have affected the prevalence found?

Response: Recruiting participants by invitation might affect validity of the finding from the study because characteristics of people who were willing or not willing to participate the study might be different such as socioeconomic status. The differences in these characteristics might be associated with the prevalence of intestinal parasitic infections and might deviate the results from the truth. This point has been discussed in the discussion part (see Section 4.1, lines 540-544).

  1. Line 116: “and Cryptosporidium by ELISA”; however, nothing is mentioned about it in the manuscript.

Response: We have added the results of Cryptosporidium identified by ELISA in the results (please see lines 283-285 and Table 2).

  1. For the 2.4.5 section, more details should be included.

Response: Drug susceptibility method was described in more details (see Section 2.5.5, lines 189-204).

  1. Table 1: please correct the number of total participants.

Response: Corrected.

  1. Is there any specific reason that could explain the significant higher rate of intestinal parasitic infections found in the children from the Border Patrol Police United Bank Bangkok School?

Response: We have explored the difference in some factors (e.g., age, sex, weight, and height) that might be related with prevalence of intestinal parasitic infections. However, these factors were not significantly different among the 3 schools included in our study.

  1. Line 195: please change “9.53%” to “3.03%”.

Response: Corrected.

  1. The prevalence of Enterobious vermicularis was 0.61%; however, this prevalence could have been much higher if a Graham Test had been performed, right?

Response: This is the limitation of our study that the helminthic infection was identified by onlymicroscopic examination.  Using a Graham Test can increase the chance to detect  Enterobious vermicularis. We have added this issue in the discussion part (please see lines 545-547).

  1. Lines 209-215: please indicate the p-values. It would also be appropriate to indicate in some way the statically significant cases in Table 3.

Response: P-values have been indicated in the results (please see lines 306-308) and Table 3.

  1. A discussion of the results obtained in relation to genotyping is missing.

Response: Discussion of genotyping results was added (please see lines 390-401).

  1. Lines 374-375: “the majority of our Giardia isolates seemed to be resistant to mebendazole as IC50 values in 30% isolates were higher than the reference isolate”. 30% is not the majority; on the other hand, are there other possible reasons that could explain these results?

Response: Since WB Clone C6 isolate is resistant to some drug but not all 3 drugs tested. We reconsider to remove the WB Clone C6 isolate from drug susceptibility part to prevent misinterpretation of drug susceptibility. The data was reanalysed.

  1. Line 383: “which may contribute to the development of drug resistance”. Please indicate references related to resistance to mebendazole/benzimidazoles.

Response: Drug susceptibility data was reanalysed. The context is changed based on the reanalysis.

  1. Line 387: “In other words, the Giardia isolates demonstrated an increased resistance to metronidazole”. This reviewer thinks that the fact that the IC50 values of metronidazole were relatively high and that it had less antiparasitic effect than the other drugs does not mean necessarily that the isolates studied show an increased resistance. Are there other possible reasons that could explain these results?

Response: Drug susceptibility data was reanalysed. The context is changed based on the reanalysis.

  1. Authors should include a paragraph with future directions.

Response: Future directions were added (lines 518-524).

  1. Articles that could be consulted and cited by the authors:

Assavapongpaiboon B, Bunkasem U, Sanprasert V, Nuchprayoon S. A Cross-Sectional Study on Intestinal Parasitic Infections in Children in Suburban Public Primary Schools, Saraburi, the Central Region of Thailand. Am J Trop Med Hyg. 2018 Mar;98(3):763-767. doi: 10.4269/ajtmh.17-0240. Epub 2018 Jan 18. PMID: 29363443; PMCID: PMC5930881.

Punsawad C, Phasuk N, Bunratsami S, Thongtup K, Viriyavejakul P, Palipoch S, Koomhin P, Nongnaul S. Prevalence of intestinal parasitic infections and associated risk factors for hookworm infections among primary schoolchildren in rural areas of Nakhon Si Thammarat, southern Thailand. BMC Public Health. 2018 Sep 14;18(1):1118. doi: 10.1186/s12889-018-6023-3. PMID: 30217180; PMCID: PMC6137929.

Response: These 2 articles have been discussed in the discussion part and cited (please see lines 415-419).

Reviewer 4 Report

This paper presents a fascinating study that involves the isolation and cultivation of Giardia in the field, as well as the evaluation of its susceptibility to therapeutic drugs. However, the interpretation of the results appears to be problematic.

>L309 Most of G. lamblia found in our study were Giardia TPI B and had a high probability of being resistant to mebendazole.

The intact strain of  WB C6 is not considered to be a resistant strain of medication drugs. As shown by *Muller et al., to generate those resistant strains (those IC50s were approx. 40 µM), the strains should be grown under increasing sub-lethal concentrations of those drugs.

In this study, all observed IC50 values were less than twice that of the intact WB C6 strain. Notably, in the case of Metronidazole, the differences in IC50 values between WB C6 and other strains isolated did not evident any lower sensitivity.

This paper includes an apparent misinterpretation regarding drug resistance. Although the results of the drug susceptibility assessment revealed that Giardia strains isolated from the field did not retain any drug resistance, authors challenge to say that  “most of G. lamblia found in this study had a high probability of being resistant to mebendazole", and thus the discussions and conclusions regarding drug resistance in this paper should be thoroughly revised.

*Müller J, Sterk M, Hemphill A, Müller N. Characterization of Giardia lamblia WB C6 clones resistant to nitazoxanide and to metronidazole. J Antimicrob Chemother. 2007 Aug;60(2):280-7. doi: 10.1093/jac/dkm205. Epub 2007 Jun 8. PMID: 17561498.

# Other comments

>Reference 2 Organization WH. Soil-transmitted helminth infections. Fact sheet, updated January 2017. 2017.

 Revise the "Organization WH." ->"World Health Organization."

>Reference 12 Davids BJ, Gillin FD. Methods for Giardia Culture, Cryopreservation, Encystation, and Excystation In Vitro.

This is a chapter of the original book [Giardia: A Model Organism]
Pages and detailed book information are required.

Giardia A Model Organism
Lujan, Hugo D. editor. ; Svärd, Staffan. editor.
1st ed. 2011., Vienna : Springer Vienna : Imprint: Springer, 2011

>L194 Trichomonas hominis and Table 2 also.
Pentatrichomonas hominis (syn. Trichomonas hominis) might be better.

>L196 (Entamoeba spp).  and Table 2 also.
"spp" is abbreviation of species, therefore, it should be shown as "spp." (not italic also).

 >Figure 1. Growth phase of G. lambia WB-C6 strain.

G.lambia -> G. lamblia

Figure 1 displays the growth phase of the G. lamblia WB-C6 strain. However, it may be not easy to comprehend since the X-axis does not show the time scale, which is crucial for following the growth phase accurately. Using multiple sets of lines based on the initial numbers of trophozoites would make the graph more understandable.

Author Response

Response to Reviewer 4’s comments

This paper presents a fascinating study that involves the isolation and cultivation of Giardia in the field, as well as the evaluation of its susceptibility to therapeutic drugs. However, the interpretation of the results appears to be problematic.

1. L309 Most of G. lamblia found in our study were Giardia TPI B and had a high probability of being resistant to mebendazole. The intact strain of  WB C6 is not considered to be a resistant strain of medication drugs. As shown by *Muller et al., to generate those resistant strains (those IC50s were approx. 40 µM), the strains should be grown under increasing sub-lethal concentrations of those drugs. In this study, all observed IC50 values were less than twice that of the intact WB C6 strain. Notably, in the case of Metronidazole, the differences in IC50 values between WB C6 and other strains isolated did not evident any lower sensitivity. This paper includes an apparent misinterpretation regarding drug resistance. Although the results of the drug susceptibility assessment revealed that Giardia strains isolated from the field did not retain any drug resistance, authors challenge to say that  “most of G. lamblia found in this study had a high probability of being resistant to mebendazole", and thus the discussions and conclusions regarding drug resistance in this paper should be thoroughly revised.
*Müller J, Sterk M, Hemphill A, Müller N. Characterization of Giardia lamblia WB C6 clones resistant to nitazoxanide and to metronidazole. J Antimicrob Chemother. 2007 Aug;60(2):280-7. doi: 10.1093/jac/dkm205. Epub 2007 Jun 8. PMID: 17561498.
Response: Since WB Clone C6 isolate is resistant to some drug but not all 3 drugs tested. We reconsider to remove the WB Clone C6 isolate from drug susceptibility part to prevent misinterpretation of drug susceptibility. The data was reanalysed to compare IC50 across the 10 Giardia isolates for each drug tested. The results in both abstract and main manuscript have been revised accordingly.

  1. Reference 2 Organization WH. Soil-transmitted helminth infections. Fact sheet, updated January 2017. 2017.
    Revise the "Organization WH." ->"World Health Organization."

Response: Revised.

  1. Reference 12 Davids BJ, Gillin FD. Methods for Giardia Culture, Cryopreservation, Encystation, and Excystation In Vitro.
    This is a chapter of the original book [Giardia: A Model Organism]
    Pages and detailed book information are required.
    Giardia A Model Organism
    Lujan, Hugo D. editor. ; Svärd, Staffan. editor.
    1st ed. 2011., Vienna : Springer Vienna : Imprint: Springer, 2011

Response: Revised.
4. L194 Trichomonas hominis and Table 2 also.
Pentatrichomonas hominis (syn. Trichomonas hominis) might be better.

Response: Revised.

  1. L196 (Entamoeba spp).  and Table 2 also.
    "spp" is abbreviation of species, therefore, it should be shown as "spp." (not italic also).

Response: Revised.
 6. Figure 1. Growth phase of G. lambia WB-C6 strain.

G.lambia -> G. lamblia

Figure 1 displays the growth phase of the G. lamblia WB-C6 strain. However, it may be not easy to comprehend since the X-axis does not show the time scale, which is crucial for following the growth phase accurately. Using multiple sets of lines based on the initial numbers of trophozoites would make the graph more understandable.

Response: We consider your point of concern. However, changing the X-axis to the time scale (24, 48, and 72 hrs) will alter the analysis of correlation factor (r2) at each number of cell seeding due to the small variable of time scale (3 values). The reanalyzed value of r2 will affect the optimal growth condition established in this study. However, we followed the published method for finding optimal growth phase of G. lamblia WB-C6 strain in the following article. This figure has been removed from the main manuscript and moved to Figure S2.

Hahn J, Seeber F, Kolodziej H, Ignatius R, Laue M, Aebischer T, et al. High Sensitivity of Giardia duodenalis to Tetrahydrolipstatin (Orlistat) In Vitro. PLOS ONE. 2013;8(8):e71597. doi: 10.1371/journal.pone.0071597

Round 2

Reviewer 1 Report

Thank you very much for responding to my observations. However, there are still some points that need to be revised.

1. Line 60-61: “Currently, eight assemblage (A-H) of G. lamblia have been identified, with assemblages A and B being associated with human infections and the others with different hosts.”. The Giardia lamblia assemblages have a high zoonotic potential. The sentence suggests that the assemblages A and B are only found in humans and the C-H in any other animal type. Please rewrite the sentence.

2. Line 62-63: “Our objective was to determine the prevalence of G. lamblia assemblage as it was the most prevalent parasite identified.” This information is included in the objective at the end of the introduction. Please remove.

3. Line 63: “There are few data available for the current state for 63 G. lamblia in Thailand.” Include the reference.

4. Line 64-66: “The inclusion of G. lamblia detection and genotyping by real time PCR can provide valuable epidemiological information and potentially enhance our understanding of G. lamblia transmission route.” Not only real-time PCR. Other genotyping methodologies also fulfill this function. Remove the methodology in the sentence.

5. Line 66-67: “In addition, lacking of the approved vaccine and effective drugs against Giardia infection may contribute to drug resistance.” Remove this sentence.

6. After citing the complete name of the species, the genus should be abbreviated throughout the text. Review all text. (Example: Ascaris lumbricoides -> A. lumbricoides; Trichuris trichiura -> T. trichiura; Enterobius vermicularis -> E. vermicularis.)

7. Line 344-346 “This indicates that G. lamblia isolates in this study were more susceptible to mebendazole than to metronidazole 17 times and they were more susceptible to furazolidone than to metronidazole 7 times.” The sentence is repetitive. Remove it.

8. Line 398-401: “The samples that were not typeable could potentially be of other assemblages that were not tested for, specifically, assemblage E which is commonly associated with grazing or herding animals but was detected in 2016 in humans”. The assemblage E in human has several reports. Rewrite the sentence.

9. Line 467-469: Identification of non-pathogenic parasites is important because it highlights fecal contamination. This information is important to include.

10. The conclusion needs to be rewritten. It must contain the main findings responding to the objectives proposed in the study.

Minor editing of English language required.

Author Response

Response to Reviewer 1

Thank you very much for responding to my observations. However, there are still some points that need to be revised.

  1. Line 60-61: “Currently, eight assemblage (A-H) of G. lamblia have been identified, with assemblages A and B being associated with human infections and the others with different hosts.”. The Giardia lamblia assemblages have a high zoonotic potential. The sentence suggests that the assemblages A and B are only found in humans and the C-H in any other animal type. Please rewrite the sentence.

Response: This sentence has been revised.

  1. Line 62-63: “Our objective was to determine the prevalence of G. lamblia assemblage as it was the most prevalent parasite identified.” This information is included in the objective at the end of the introduction. Please remove.

Response: This sentence has been deleted.

  1. Line 63: “There are few data available for the current state for 63 G. lamblia in Thailand.” Include the reference.

Response: The references have been added.

  1. Line 64-66: “The inclusion of G. lamblia detection and genotyping by real time PCR can provide valuable epidemiological information and potentially enhance our understanding of G. lamblia transmission route.” Not only real-time PCR. Other genotyping methodologies also fulfill this function. Remove the methodology in the sentence.

Response: The methodology has been removed from the sentence.

  1. Line 66-67: “In addition, lacking of the approved vaccine and effective drugs against Giardia infection may contribute to drug resistance.” Remove this sentence.

Response: This sentence has been removed.

  1. After citing the complete name of the species, the genus should be abbreviated throughout the text. Review all text. (Example: Ascaris lumbricoides -> A. lumbricoides; Trichuris trichiura -> T. trichiura; Enterobius vermicularis -> E. vermicularis.)

Response: Revised.

  1. Line 344-346 “This indicates that G. lamblia isolates in this study were more susceptible to mebendazole than to metronidazole 17 times and they were more susceptible to furazolidone than to metronidazole 7 times.” The sentence is repetitive. Remove it.

Response: The sentence has been removed.

  1. Line 398-401: “The samples that were not typeable could potentially be of other assemblages that were not tested for, specifically, assemblage E which is commonly associated with grazing or herding animals but was detected in 2016 in humans”. The assemblage E in human has several reports. Rewrite the sentence.

Response: This sentence has been rewritten.

  1. Line 467-469: Identification of non-pathogenic parasites is important because it highlights fecal contamination. This information is important to include.

Response: This sentence has been added in the discussion part.

  1. The conclusion needs to be rewritten. It must contain the main findings responding to the objectives proposed in the study.

Response: The conclusion has been revised.

Reviewer 2 Report

Add a suitable reference for lines 44–46.

Add a suitable reference for lines 47–50.

Add a suitable reference for lines 55–57.

Add a suitable reference for lines 60–61.

Add suitable references for lines 63–67.

Add suitable references for headings 2.4, 2.5.1, 2.5.2, and 2.5.3.

What kind of ELISA kit did you use? (direct ELISA, indirect ELISA, etc.)

μl should be μL. Check and correct it throughout the text.

The half maximal inhibitory concentrations (IC50) was determined. (Is it time-dependent or dose-dependent? explain it)

increasing resistance or tolerance to commonly used anti-parasitic agents (mention the name of the drugs).

Do not present the same data in text, table, and/or figure.

In Figure 2, label the axis of each graph.

Line 331: Check and correct the spelling of species.

In the discussion, do not waste entire sentences restating your results. Lines 383–389 should be part of the results, not the discussion.

Organize the discussion to address each of the experiments or studies for which you presented results. Rather than starting the discussion of genotyping, start the discussion with the compression of prevalence results, then association with nutritional status, excystation of G. lamblia cysts, drug susceptibility tests of G. lamblia isolates, and then G. lamblia genotypes.

Replace reference No. 10 with a more suitable reference.

Ascaris lumbricoides, Trichuris trichiura, and hookworms were the most prevalent helminths among these parasites, (among which parasites? Rephrase this sentence.)

The format of references is not the same; please check and correct according to journal format.

Extensive editing of English language required

Author Response

Response to Reviewer 2 comments

  1. Add a suitable reference for lines 44–46.

Response: Added (please see reference no. 6).

  1. Add a suitable reference for lines 47–50.

Response: Added (please see reference no. 7).

  1. Add a suitable reference for lines 55–57.

Response: Added (please see reference no. 12,13)

  1. Add a suitable reference for lines 60–61.

Response: Added (please see reference no. 15).

  1. Add suitable references for lines 63–67.

Response: Added (please see reference no. 16)

  1. Add suitable references for headings 2.4, 2.5.1, 2.5.2, and 2.5.3.

Response: Added.

  1. What kind of ELISA kit did you use? (direct ELISA, indirect ELISA, etc.)

Response: The information of ELISA kit has been added (Please see lines 180-182).  

  1. μl should be μL. Check and correct it throughout the text.

Response: Checked and corrected.

  1. The half maximal inhibitory concentrations (IC50) was determined. (Is it time-dependent or dose-dependent? explain it)

Response: It is dose-dependent (please see line 214).

  1. Increasing resistance or tolerance to commonly used anti-parasitic agents (mention the name of the drugs).

Response: Drug names are mentioned (please see line 73).

  1. Do not present the same data in text, table, and/or figure.

Response. We have revised the results for not duplicating the information in text, figure, and table.

  1. In Figure 2, label the axis of each graph.

Response: The X-axis has been labeled.

  1. Line 331: Check and correct the spelling of species.

Response: corrected.

  1. In the discussion, do not waste entire sentences restating your results. Lines 383–389 should be part of the results, not the discussion.

Response: We have deleted this paragraph from the discussion.

  1. Organize the discussion to address each of the experiments or studies for which you presented results. Rather than starting the discussion of genotyping, start the discussion with the compression of prevalence results, then association with nutritional status, excystation of G. lamblia cysts, drug susceptibility tests of G. lamblia isolates, and then G. lamblia genotypes.

Response: We have reorganized the sequence of discussion according to your suggestion.

  1. Replace reference No. 10 with a more suitable reference.

Response: Reference no. 10 has been replaced (please see reference no. 14).

  1. Ascaris lumbricoides, Trichuris trichiura, and hookworms were the most prevalent helminths among these parasites, (among which parasites? Rephrase this sentence.)

Response: We apologize for our mistake. The phrase “among these parasites” has been deleted.

  1. The format of references is not the same; please check and correct according to journal format.

Response: Checked and corrected.

Response to Reviewer 2 comments

  1. Add a suitable reference for lines 44–46.

Response: Added (please see reference no. 6).

  1. Add a suitable reference for lines 47–50.

Response: Added (please see reference no. 7).

  1. Add a suitable reference for lines 55–57.

Response: Added (please see reference no. 12,13)

  1. Add a suitable reference for lines 60–61.

Response: Added (please see reference no. 15).

  1. Add suitable references for lines 63–67.

Response: Added (please see reference no. 16)

  1. Add suitable references for headings 2.4, 2.5.1, 2.5.2, and 2.5.3.

Response: Added.

  1. What kind of ELISA kit did you use? (direct ELISA, indirect ELISA, etc.)

Response: The information of ELISA kit has been added (Please see lines 180-182).  

  1. μl should be μL. Check and correct it throughout the text.

Response: Checked and corrected.

  1. The half maximal inhibitory concentrations (IC50) was determined. (Is it time-dependent or dose-dependent? explain it)

Response: It is dose-dependent (please see line 214).

  1. Increasing resistance or tolerance to commonly used anti-parasitic agents (mention the name of the drugs).

Response: Drug names are mentioned (please see line 73).

  1. Do not present the same data in text, table, and/or figure.

Response. We have revised the results for not duplicating the information in text, figure, and table.

  1. In Figure 2, label the axis of each graph.

Response: The X-axis has been labeled.

  1. Line 331: Check and correct the spelling of species.

Response: corrected.

  1. In the discussion, do not waste entire sentences restating your results. Lines 383–389 should be part of the results, not the discussion.

Response: We have deleted this paragraph from the discussion.

  1. Organize the discussion to address each of the experiments or studies for which you presented results. Rather than starting the discussion of genotyping, start the discussion with the compression of prevalence results, then association with nutritional status, excystation of G. lamblia cysts, drug susceptibility tests of G. lamblia isolates, and then G. lamblia genotypes.

Response: We have reorganized the sequence of discussion according to your suggestion.

  1. Replace reference No. 10 with a more suitable reference.

Response: Reference no. 10 has been replaced (please see reference no. 14).

  1. Ascaris lumbricoides, Trichuris trichiura, and hookworms were the most prevalent helminths among these parasites, (among which parasites? Rephrase this sentence.)

Response: We apologize for our mistake. The phrase “among these parasites” has been deleted.

  1. The format of references is not the same; please check and correct according to journal format.

Response: Checked and corrected.

Reviewer 3 Report

I thank the authors for their responses to my comments; however, there are still some minor points that need to be addressed before accepting the manuscript:

Once the species are named, use the abbreviated genus in subsequent mentions (for example, lines 287, 289, 309, 448, 449…).

Lines 66-67: “In addition, lacking of the approved vaccine and effective drugs against Giardia infection may contribute to drug resistance”. I think that this sentence is confusing.

Lines 70-71: “to commonly used anti parasitic agents, which are the drug of choice for Giardia therapy”. Please rewrite.

Line 80: “to monitor distribution of drug resistance in Giardia in Thailand”. Please delete or rewrite this sentence according to the changes made in the manuscript.

Lines 449-450: “the presence of Ascaris lumbricoides or Enterobius vermicularis and being underweight, stunted, or wasted”. Please rewrite this sentence.

Lines 533-534: “With the limited…in this country”. Please rewrite this sentence according to the changes made in the manuscript.

Lines 542-543: “characteristics such as socioeconomic status, of people who were willing or not willing to participate the study might be different” change to “characteristics of people who were willing or not willing to participate the study might be different such as socioeconomic status”.

Minor editing of English language required.

Author Response

Response to Reviewer 3

I thank the authors for their responses to my comments; however, there are still some minor points that need to be addressed before accepting the manuscript:

  1. Once the species are named, use the abbreviated genus in subsequent mentions (for example, lines 287, 289, 309, 448, 449…).

Response: Corrected.

  1. Lines 66-67: “In addition, lacking of the approved vaccine and effective drugs against Giardia infection may contribute to drug resistance”. I think that this sentence is confusing.

Response: This sentence has been rewritten (please see lines 67-68).

  1. Lines 70-71: “to commonly used anti parasitic agents, which are the drug of choice for Giardia therapy”. Please rewrite.

Response: This sentence has been rewritten (please see lines 72-73).

  1. Line 80: “to monitor distribution of drug resistance in Giardia in Thailand”. Please delete or rewrite this sentence according to the changes made in the manuscript.

Response: This sentence has been rewritten (please see lines 83-84).

  1. Lines 449-450: “the presence of Ascaris lumbricoides or Enterobius vermicularis and being underweight, stunted, or wasted”. Please rewrite this sentence.

Response: This sentence has been revised (please see lines 428-429).

  1. Lines 533-534: “With the limited…in this country”. Please rewrite this sentence according to the changes made in the manuscript.

Response: This sentence has been rewritten (please see lines 523-525).

  1. Lines 542-543: “characteristics such as socioeconomic status, of people who were willing or not willing to participate the study might be different” change to “characteristics of people who were willing or not willing to participate the study might be different such as socioeconomic status”.

Response: Revised (please see lines 532-533).

Reviewer 4 Report

This paper is excellent, both in terms of the research and the paper's description. The revised discussion of drug resistance now appears to be appropriate.

As a minor correction, please change the following:
Table 3: Blastocystis Hominis -> Blastocytis hominis

Author Response

Response to Reviewer 4 comments

This paper is excellent, both in terms of the research and the paper's description. The revised discussion of drug resistance now appears to be appropriate.

  1. As a minor correction, please change the following: Table 3: Blastocystis Hominis -> Blastocytis hominis

Response: Corrected.

Round 3

Reviewer 2 Report

Accept in present form

Minor editing of English language required

Author Response

  • Minor editing of English language required.

Response: English language has been edited.